# Pregnant Women’s Views Regarding Maternity Facility-Based Delivery at Primary Health Care Facilities in the Province of KwaZulu-Natal in South Africa

**DOI:** 10.3390/ijerph20156535

**Published:** 2023-08-06

**Authors:** Puseletso Ruth Mlotshwa, Maureen Nokuthula Sibiya

**Affiliations:** 1Faculty of Health Sciences, Durban University of Technology, Durban 4000, South Africa; prmlotshwa@gmail.com; 2Division of Research, Innovation and Engagement, Mangosuthu University of Technology, Umlazi 4031, South Africa

**Keywords:** acceptance, antenatal care, facility-based delivery, South Africa

## Abstract

For women giving birth, every moment of delay in receiving skilled care significantly increases the risks of stillbirth, neonatal and maternal death. More than half of all births in developing countries, including South Africa, take place outside a health facility and without skilled birth attendants. Therefore, this has made it difficult to achieve the Sustainable Development Goals of global reduction in maternal mortality, which is a key health challenge globally, especially in developing countries and sub-Saharan Africa in particular. The study aimed to explore and describe the views of pregnant women regarding facility-based delivery. Focus group discussions were used to gather information from pregnant women. Information was collected from six groups of pregnant women who had delivered babies at the primary health care facilities in the past 5 years. Results showed several factors associated with the failure to use institutional delivery services, such as the lengthy distance from the health care facility, lack of transport, lack of transport fare, shortages of skilled staff, failure to disclose pregnancy, cultural and religious beliefs, and staff attitudes.

## 1. Introduction

Antenatal care (ANC) is pivotal in improving maternal and child health [1]. It is also an important determinant of maternal and perinatal mortality and ANC attendance is an essential component of maternal health care on which the health of mothers and newborns depend [1,2,3,4,5]. ANC, along with skilled delivery care, is a key element of the package of services aimed at improving maternal and newborn health. Skilled attendance throughout pregnancy from inception to postnatal stage ensures safe birth, reduces potential and actual complications, and increases the survival of both the mother and newborn [6,7]. However, it is estimated that only 50% of pregnant women in the world have access to skilled delivery [8], with sub-Saharan Africa (SSA) having far fewer numbers of women able to deliver at health facilities [5,7,9]. For instance, a comparative study conducted in Nigeria and Ethiopia found that 85% of pregnant women deliver at home in Ethiopia, whereas one-fifth of the deliveries in Nigeria occur without anyone present [7]. There is empirical evidence of inequities in health service delivery and utilization between urban and rural settings and across various racial groups, and by inference, across the nine provinces, which differ in terms of socio-demographics and geography [7,9]. According to these findings, maternal and child healthcare is not reaching everyone in rural SSA including South Africa. Variations are due to factors like poverty, low levels of education and unemployment, among many others. This is supported by the findings of the study conducted by Bomela, which revealed that the majority of deaths occurred among those whose educational status was unspecified (46.2%) and lowest among those with no education, primary school education and university education, probably due to their health seeking behaviours [10]. Maternal deaths have steadily declined in South Africa since 2009, from a high of 2275 deaths to a low of 1097 deaths in 2015. This decline could be attributed to improvements in HIV treatment with the extensive provision of ARVs to pregnant women, which affects mostly young women in South Africa [10]. The decline in HIV is a significant contribution towards the Sustainable Development Goal (SDG), indicator: 3.3 in the fight against the AIDS epidemic [11]. Furthermore, South Africa continues to improve its policies for further reduction of maternal mortality rates by establishing and implementing strategies to improve maternal health [12]. According to Yaya et al. [7], women in rural areas in Nigeria and Ethiopia regarded facility delivery as uncustomary and complained of long distances to travel to health facilities, whereas urban women complained about the poor quality of care and high cost of hospital delivery.

Numerous challenges that obstruct safe maternal health care were identified by the National Department of Health of South Africa and are indicated in the National Committee on Confidential Enquiries into Maternal Deaths (NCCEMD) 2018 Report as needing immediate attention [2]. Empirical studies in SSA, including South Africa, have identified socio-cultural factors, including maternal factors, ANC-related factors, facility-related factors (accessibility of quality services) and macro-level factors as barriers to the utilization of facility-based delivery (FBD) [4,7,9,13]. According to Dzomeku [14] and Wassihun et al., [15], fear of maltreatment, as well as the negative attitudes and behaviours of caregivers such as shouting and showing disrespect to women during delivery at health facilities feature as some of the most important barriers to FBD service utilization. Dzomeku found that women after delivery felt dissatisfied with the quality of care received at the health facilities [14]. The healthcare system in South Africa is generally beset with a critical shortage of doctors, nurses and community healthcare workers; health facilities are either in disrepair or ill-equipped, equipment damaged or unavailable, together with dysfunctional emergency medical services, especially interfacility transport between and within provinces [2]. Though every country in the SSA has experienced the scourge of the unsupervised births, South Africa and KwaZulu-Natal (KZN) province, in particular, seem to be the epicentre of the challenge. Most maternal deaths in the 10–24 years age group occurred in KZN (36.2%) [4]. According to the 2016 South Africa Demographic and Health Survey (SADHS), about 94% of pregnant women in KZN received ANC during pregnancy with as many as 76% attending at least four ANC visits, yet less than 55% of these women, mostly in the rural communities, received skilled attendance at available health facilities, especially women aged 35–49 years [16].

In 2005, the South African Department of Health adopted and modified the Focused Antenatal Care (FANC) Model to suit the South African circumstances and referred to it as the Basic Antenatal Care (BANC) approach [17]. BANC is an approach that was implemented in 2008 as a quality improvement strategy to be used by all the public health institutions in South Africa to provide ANC services and is the recommended minimum level of ANC that every pregnant woman should receive [18]. According to Pattinson, every site where pregnant women make contact with health services should provide daily BANC services so that the first ANC consultation takes place as soon as the pregnancy has been confirmed or, at the very last, the first time that a pregnant woman visits a health facility [19].

Moodley et al. reported that around 60% of all maternal deaths in South Africa between 2014 and 2016 were attributed to poor quality of care and were preventable [12]. Therefore, this has made it difficult to achieve the SDGs of the global reduction in maternal mortality, which is a key health challenge globally, especially in developing countries and SSA in particular [19]. Therefore, the aim of this study was to explore and describe the views of pregnant women regarding maternity FBD to promote the better acceptance of delivering at PHC facilities in KZN.

## 2. Materials and Methods

### 2.1. Study Site and Context

This study used a qualitative, exploratory and descriptive research design to explore and describe the views of pregnant women regarding FBD. Data was collected in two Community Health Centres (CHCs) that are located in uMzinyathi District and uThukela District. The sites are referred to CHC A and B. CHC B was officially opened on 15 November 2011; the first CHC in uThukela District. It is situated at uMnambithi Municipality in Ladysmith, and its key priority is to serve the population that resides in Indaka Local Municipality and uMnambithi Municipality. CHC B is a referral point for nine clinics, namely KwaMteyi, Gcinalishona, eZakheni (E), eZakheni NO_2_ (C), Sigweje, Ekuvukeni, Rockliff, Limehill, and Sahlumbe Clinic. CHC B provides a range of health care services in its quest to deliver optimal health care services that are integrated with the district health system within its catchment areas, namely sexual and reproductive health growth, ANC, labour, delivery and postnatal care (PNC), immunization and child growth, among others.

### 2.2. Data Collection

A purposive sampling technique was used to select a total of 14 participants from the health facilities under study. The participants under consideration for inclusion in this study included women who were pregnant and had previous deliveries in the past 5 years. Nine pregnant women were from CHC-A and five pregnant women from CHC-B. The pregnant women were divided into six homogenous groups. Two to three participants were purposively selected from each homogenous group, in order to promote a comfortable group dynamic for them to share their thoughts freely about the phenomenon under investigation [20,21]. The possible obstacles to the homogeneity of the groups were age and race but effects (if any) would be insignificant, since participants were of the same generational group between 20 years and 45 years. The number of discussions held was determined by data saturation. All pregnant women who participated in this study had a previous delivery. It was deemed necessary to assess pregnant women’s views about facilitators and challenges of facility-based delivery because they can provide the required information based on their experience and also for ease of access. Table 1 presents the description of the demographic characteristics of participants. Focus group discussions (FGDs) were used to collect data. Data collection occurred between June and November 2020 in the ANC unit. The ages of pregnant women ranged between 25 and 37 years of age. Their highest level of education was high school grades, with 6 primary grades being the lowest level of education.

A FGD guide with semi-structured questions was used to gather data from the pregnant women. The guide was generated from the findings of the literature review, which guided and directed the discussions. The guide was developed in English and translated into isiZulu as it is the main language spoken and understood by participants in the study area. To ensure accuracy of the translated FGD guide, the translation was performed by a professional translator. To establish the views of pregnant women regarding maternity FBD, the following main questions were asked:


*“What do you know about facility-based delivery?”*



*“Why do you think pregnant women do not utilize health facilities for childbirth?”*



*“What do you think are the facilitating conditions that promote facility-based delivery?”*


This was followed by prompts and probing questions to encourage the participants to elaborate or further expound on their responses.

### 2.3. Data Analysis

The data obtained from the FDG were analysed using the framework analysis technique [20]. This is a qualitative thematic analysis method that involves a systematic and rigorous process of classifying and organising qualitative data into themes and sub-themes. The five stages of the framework analysis technique were applied as follows: (1) transcribing the data, (2) familiarising oneself by immersing within the data to obtain detailed insights of the whole data set, (3) coding, (4) charting by linking codes to form overarching sub-themes and major themes and (5) mapping and interpreting the data.

### 2.4. Ethical Considerations

Ethical clearance to conduct this study was obtained from the Durban University of Technology’s Institutional Research Ethics Committee (reference number: 131/19) on 9 October 2019. Approval was also granted by the Province of KZN Department of Health on 27 September 2019 (reference number: KZ_201909_028). Participants provided written consent for their voluntary participation in the study. To protect the identities of the participants, in this paper, we have anonymised their identities by assigning them pseudonyms. The researcher ensured the ethics of the study by strictly applying the following ethical principles: the choice to participate, justice, veracity, beneficence, anonymity, and confidentiality.

## 3. Results

Four main themes emerged inductively from the data sources to describe the participants’ views on FBD. The main themes included: (a) conceptualisation of FBD; (b) benefits of FBD; (c) hindrances to FBD; and (d) facilitating conditions to FBD.

### 3.1. Theme 1: Conceptualisation of Facility-Based Delivery

Two sub-themes emerged as participants described the concept of FBD to demonstrate their level of knowledge, namely: skilled delivery and health care facility delivery.

#### 3.1.1. Skilled Delivery

Skilled delivery was one of the major concepts that continuously emerged from the data as a feature of FBD. Pregnant women conceptualised FBD as a delivery supervised by trained personnel who possess the necessary skills and knowledge to reduce potential complications due to pregnancy and childbirth. The statements below explain this further:


*“The facility-based delivery is conducted at the clinics by health professionals who also make sure that all hidden diseases are diagnosed early when pregnant women attend the ANC.”*
[P#2].


*“Eh…the clinic helps us (pregnant women) to be safe and child to be right. This is where pregnant women are assessed to find problems during pregnancy and be treated or managed earlier or referred for further management.”*
[P#3].

#### 3.1.2. Health Care Facility Delivery

The participants conceptualised the health care facility delivery as an attribute of FBD. FBD was described as delivering at a clinic or health centre that is within the reach of the pregnant women in the community. They explained that emergency referrals are easily carried out to first level hospitals to prevent death in cases of excessive bleeding and other related dangers during deliveries. The following are selected excerpts from the different groups of participants which highlight this further:


*“It is a health facility where everybody is treated to be well and making sure that there is no loss of babies or mothers’ lives.”*
[P#2].


*“It is the health facility which is closer to our homes and is easy to access when one is in labour or is ill.”*
[P#1].

### 3.2. Theme 2: Benefits of Facility-Based Delivery

From the data sources, three sub-themes emerged as benefits of FBD: safe delivery; free delivery services and a robust referral system.

#### 3.2.1. Safe Delivery

Pregnant women concur with each other that safety is guaranteed for them when delivering at health facilities run by knowledgeable, skilled, and experienced health personnel. This is noted in the excerpts below:


*“Yes, the benefit of facility-based delivery is that pregnant women are delivered by well-trained midwives who have all the skills and knowledge without sustaining complications like bleeding, tears and even death.”*
[P#5].


*“Eh, it really benefits pregnant women to deliver at a clinic because the babies that they deliver are bouncy and healthy if they have been attending the ANC and if problems arise the mothers are or with the babies then transferred to hospital for further management.”*
[P#6].

These narratives suggest that women prefer to deliver at health facilities based on the expertise and competence of the health providers. According to the pregnant women, compassionate and respectful care coupled with good quality obstetric care motivated them to deliver at such facilities. Another pregnant woman narrated that health facilities were safe havens as they received full attention and individualized care during delivery. This is illustrated from the narrative below:


*“Delivering at a health facility is safe. We get full attention with our unborn babies during the antenatal visits and during labour as well.”*
[P#10].


*“Oh yes! There are a lot of benefits, as we women are assessed early for any pregnancy related problems or illnesses and treated early before complications arise. We are then delivered safely by well trained and knowledgeable nurses/midwives.”*
[P#8].

Trust and confidence in the health providers’ expertise and competence resulting in good quality obstetric care increases patronage of FBD. According to the pregnant women, delivering at a health facility helps in the early detection of health problems before labour and delivery. This is attributed to the proper observation and assessment of pregnant women. The statement below explains this further:


*“If a woman is sick like having a high blood pressure, the nurses quickly diagnose her and prescribe treatment and further management.”*
[P#11].

The participants expressed their excitement that all pregnant women, whether poor or wealthy, are able to access the health facility without paying a cent (free service), even if a woman undergoes a caesarean section. They further argued that undergoing a caesarean section is a lifesaving operation that cannot be performed by the Traditional Birth Attendant (TBA), as this procedure requires highly skilled medical and health professionals with special expertise, thus preventing maternal and neonatal mortality. The following are selected excerpts that highlight this:


*“First and the most exciting experience is that the services are free of charge, and we don’t even pay for the medication that is prescribed for us for any illness. “We also don’t pay for delivery even if it is a Caesarean section.”*
[P#12].

One of the participants pointed out that every pregnant woman is welcome and treated equally with respect at the health facility. Health education provided to pregnant women helps to provide skills and knowledge in caring for the babies after delivery.

The above excerpts show that FBD is associated with improved outcomes regarding delivery services for pregnant women. Considering these perceived benefits, pregnant women are motivated to visit health facilities for delivery services.

#### 3.2.2. Free Delivery Services

The participants reported that they received free services relating to pregnancy and delivery, which helped in reducing maternal and child mortality. They further appreciated that FBD was free, readily available, and accessible to pregnant women in labour because it was a priority of the government to reduce maternal and child mortality in the country. The quote below supports this:


*“Eh… facility-based delivery is easily accessible to many pregnant women; it is nearby and is usually built where many people live. The care given to all pregnant women and women in labour is free of charge.”*
[P#2].

#### 3.2.3. Robust Referral System

The facility-based delivery system has strengthened the referral system. Women with complicated delivery procedures are referred to the next level of care for proper and competent care. Participants reported that this facilitates proper implementation of the referral system within and among health facilities in the country. One pregnant woman narrated her ordeal when she was diagnosed with breech presentation and was therefore referred to a level 1 facility where she received the best care. A level 1 facility is the provincial level of hospitals where patients are usually referred to from the clinics. Receiving the best care motivated her to continuously seek delivery services from the facility. This is illustrated in the narrative below:


*“It was good, and I am grateful because during the antenatal care the midwives were able to diagnose that my baby was a breech presentation. They then referred me to the level 1 health facility and at level 1 facility I received the best care. I am very grateful for the care I received from the nurses and doctor when I was referred to level 1 facility. I thereafter delivered all my children at a health facility.”*
[P#5].

### 3.3. Theme 3: Hindrances to Facility-Based Delivery

Multiple hindrances emerged from the data analysis, which included seven sub-themes, namely: (a) long distance to health facilities; (b) resource constraints; (c) negative attitudes of staff; (d) fear associated with facility delivery; (f) cultural beliefs, (g) negligence and capricious weather (heavy rains).

#### 3.3.1. Long Distance to Health Facilities

The majority of the participants reported that long distances to health facilities affected their access to facility-based delivery services as they do not have transport of their own to take them to the health facility. They further raised concerns that unemployment makes things worse, as most of the women in rural areas are unemployed. Pregnant women in the rural areas walk long distances to access skilled delivery at health facilities. The following narratives from the participants illustrate their concerns:


*“Heh others stay far from the clinic, and it is not even easy for the ambulance to reach them, if it does it is too late.”*
[P#2].


*“It is a long distance to walk to the clinic and it is difficult to access the clinic particularly at night and when the weather is too hot or if it is raining heavily though heavy rains are rare. Consequently, some women deliver before they arrive at the health facility.”*
[P#4].

These narratives suggest that some pregnant women would prefer to deliver at home due to the long distances to health facilities. According to the pregnant women, they preferred home delivery to FBD because of the long distances to health facilities. This was intended to avoid the risks of travelling long distances to health facilities as noted in the excerpt below:


*“Some of the women stay far from the health facility and because of the long distance she has to walk or travel she then resorts to delivering at home to cut matters short.”*
[P#8].

#### 3.3.2. Resource Constraints

Adequate numbers of midwives at health facilities play a crucial role. Pregnant women reported a shortage of staff, which made them scared of health facility deliveries as they might not be well managed, attributed to many women in labour with few healthcare providers to conduct deliveries. Participants concurred that the shortage was more acute at night:


*“Yes, the shortage of staff made me scared that I might not be well attended to because of too many women in labour and yet there is not enough staff to do the job.”*
[P#4].

Another pregnant woman recounted her ordeal at a health facility where she was left unattended during labour due to the lack of staff, and she delivered alone. According to her, there were no nurses to do the observations in the PNC ward as well. She further reported that this experience affected her patronage of FBD. Below is the excerpt that describes participant’s experience:


*“I was left alone when I was in labour, and I delivered alone. After that, no nurse to do observations at the postnatal ward.”*
[P#5].

Additionally, logistical constraints, in terms of inadequate equipment and supplies in the wards, were identified by participants as one of the challenges affecting delivery services at the health facilities. This was characterized by a lack of hot water for baths post-delivery in the post-natal ward. The women described this experience as horrific, particularly in winter periods. They also reported inadequate medications to relieve pain after delivery, especially those who had undergone episiotomies. This is explained by the excerpts below:


*“The postnatal ward doesn’t have hot water for baths/showers but only cold water for the shower. This is terrible in the winter.”*
[P#6].


*“No pain tablets are given to us for after pain since childbirth is very sore especially after delivery with episiotomies.”*
[P#3].

#### 3.3.3. Negative Attitudes of Staff

Participants perceived negative attitudes of the staff (midwives) as barriers to FBD. The majority of participants reported on the staff’s rude behaviour towards them during delivery services in the maternity ward. This is illustrated by the excerpt below:


*“Nurses are rude, and they shout at us (pregnant women) when in labour. Nurses’ attitude is worse towards grand multiparous women. They continue to say that the nurses’ language affects them emotionally during delivery in the health facility.”*
[P#11].

Two participants cited their experiences to confirm this negative attitude:


*“….. staff attitude discourages us from delivering at the health facilities. Because they shout at us and make fools of us when we are in pain during labour. We feel ashamed of being belittled by nurses in front of other women.”*
[P#6].


*“Midwives had no time for us and their attitude was bad. They were talking badly to us.”*
[P#7].

#### 3.3.4. Fear Associated with Facility Delivery

Pregnant women highlighted their fears associated with facility delivery in the form of procedures such as episiotomies, the referral system and the attitude of the staff towards pregnant women. Some women mentioned that delivering at a health facility where there are no doctors exposes them to enduring extreme pain. This is expressed in the excerpts below:


*“……they say; they [pregnant women] are scared of episiotomy because the episiotomy is painful and uncomfortable. Another thing, they [pregnant women] expressed fear of being transferred to strange environment should the abnormalities occur.”*
[P#7].

The study further revealed that the world faced unprecedented healthcare challenges caused by COVID-19, one of which was FBD services. Pregnant women weighed the competing risks of COVID-19 infection versus travelling long distances to health facilities for delivery. One of the pregnant women narrated:


*“I am always afraid of coming here (clinic) because of COVID-19…….I am scared that I might contract it”*
[P#10].

#### 3.3.5. Cultural Beliefs

Indigenous cultural practices and the belief systems of pregnant women affected the patronage of FBD. The majority of the pregnant women believed that their culture forbade them from delivering at the health facilities but rather stipulated that they deliver at home either helped by their mothers-in-law or by TBAs. They [pregnant women] believed in the competence of the TBAs to produce positive delivery outcomes as indicated in the excerpt below:

*“For some of us [pregnant women], culture forbids us from delivering in the health facility…*. *In some of our families the mothers-in-law deliver us at home or sometimes the lay midwives are hired to conduct delivery.”*[P#9].

Some participants reported that home deliveries are not without complications, as some women sustain vaginal and perineal tears, and uterine ruptures with excessive bleeding and loss of lives. Some pregnant women’s cultural and religious beliefs prevented them from utilizing the health facilities for the delivery of their babies. It was further revealed that broken bottles were used at home to perform episiotomy as relevant equipment were never available for delivery purposes in such settings. Women in labour were also forced to push prematurely during delivery, which resulted in vulval swelling and difficulty in delivery. Consequently, most of them were brought to health facilities with complications that could have been prevented; this is the excerpt below, which explains how the episiotomies are performed at home:


*“Broken bottles are used for episiotomies during home delivery as there is no equipment for that purpose. Women are also forced to push prematurely during labour and this causes swelling of the vulva.”*
[P#7].

### 3.4. Theme 4: Facilitating Conditions to Facility-Based Delivery

Three sub-themes were identified, namely: good nurse-client relationship, adequate skilled staff members and availability of logistics.

#### 3.4.1. Good Nurse-Client Relationship

The interpersonal relationship between pregnant women and health personnel is an important determinant in shaping the perceptions and experiences of the former towards FBD. The majority of the pregnant women in this study reported good human relations between the nurse and expectant mothers, coupled with good nursing care that facilitated facility delivery. This is illustrated by the narrative below:


*“What promotes facility-based delivery is the good human relations between nurses and us (pregnant women), as well as the good nursing care we get from the nursing staff.”*
[P#4].

During the FGDs, the participants mentioned that attributes, such as dignified, compassionate and respectful maternity care provided at health facilities, motivated them to deliver at the various facilities. Therefore, they urged nurses to provide love and care during delivery. The following narratives from the pregnant women illustrate the good nurse-client relationship:


*“When we, at the clinics are treated with dignity and respect as human beings worth of love and care, we can use the health facilities for deliveries.”*
[P#4].


*“We need love and care from the nurses during delivery.”*
[P#5].

#### 3.4.2. Adequate Skilled Staff

Health professionals with adequate skills and experience promote FBD. A pregnant woman recounted that when a midwife diagnosed her with a breech presentation and immediately referred her to a higher facility, it saved her life and that of her baby. This is illustrated by the narrative below:


*“A midwife was able to diagnose that my baby was in a breech presentation. They then referred me to the level 1 health facility and also at level 1 health facility I received the best care.”*
[P#5].

#### 3.4.3. Availability of Logistics

The availability of pain medications and warm water for baths at the PNC unit encouraged them to deliver at hospitals. Availability of hot showers was identified as crucial, especially during the winter when the weather is cold as noted below:


*“Like at this health facility, I wish hot water could also be provided for mothers who have delivered to be able have a comfortable bath. Another thing that is important is the availability of medication whether it is for pain or any to help the mother and baby.”*
[P#4].

However, one of the pregnant women recounted a story about broken bottles being used to conduct deliveries at home by a TBA. According to the pregnant woman, equipment used at the health facilities was never available at home. Therefore, she visited a TBA after a delivery where broken bottles had been used to perform episiotomy. She described it as a very painful experience. Some of the pregnant women concurred with her. The excerpt below confirms this:


*“I also think the availability of equipment is very important unlike at home where there is no equipment and, in most instances, they use a broken bottle in making the episiotomy.”*
[P#9].

## 4. Discussion

The results of this study revealed that pregnant women displayed clear knowledge of skilled delivery as being conducted by knowledgeable nurses with special training and skills to manage childbirth and the immediate post-natal period. Pregnant women also understood that they would be delivered safely by well-trained and knowledgeable midwives at the various health facilities, as they receive individualized care during delivery and also obtain firsthand treatment during emergencies to prevent complications during pregnancy and delivery. This includes the treatment of hidden diseases during pregnancy. This study agrees with the study conducted in Ethiopia by Ombere indicating that health professionals have better skills to attend birth than TBAs [22]. Raymondville et al. and Zenbaba et al. further indicate that FBD is one of the maternal health services supported by skilled birth attendants within the health care facility, and it is recognized as an intermediation to advance maternal health that moderates complications during delivery [23,24].

The participants further highlighted some hinderances to FBD, which includes transportation costs. Although maternal and child healthcare service in South Africa is free, some participants highlighted the transportation cost being a higher cost to pay, especially if labour comes at night. This finding concurs with the studies conducted by Yaya et al., Raymondville et al. and Rahman et al. [7,23,25]. All these studies state that the cost of transportation is one of the top barriers reported for not attending facility delivery in both the urban and rural areas. Pregnant women indicated a clear awareness and knowledge of risks associated with childbirth without a skilled attendant but also mentioned poor socioeconomic conditions as most of them are unemployed. These results corroborate the findings of the study by Sibiya et al., which revealed that late attendance of ANC was common among pregnant women who are from low socio-economic backgrounds [4]. These pregnant women were all aware of the importance of delivering at a healthcare facility run by skilled and knowledgeable health professionals.

The participants reported inadequate staffing at times contributing to women delivering unattended by midwives or ending up being helped by other patients. These findings were supported by Raymondville et al. who also stated that in the study they conducted in Haiti, women reported inadequate beds and space which delayed provision of adequate care [21]. The findings of the study that was conducted in Malawi revealed multiple factors associated with delayed ANC, and these include, among others, hospital inefficiencies [3]. Interestingly, some studies have shown that large numbers of maternal and neonatal deaths occur in facilities, and that mortality rate is not reduced through FBD [26,27,28]. This is thought to be explained by facilities lacking skilled personnel and necessary equipment.

Pregnant women who participated in this study described the lack of previous physical and emotional support from the staff at the healthcare facilities as a deterrent to delivering at a health care facility. They preferred to deliver at home where they received both physical and emotional support. This statement was supported by Adatara et al. in the study they conducted in Ghana stating that women reported a lack of cultural sensitivity or respect for women’s traditional beliefs during the utilisation of skilled birth care, thus deterring them from using healthcare facilities for childbirth [29]. Cultural factors and fear of hospital deliveries were also highlighted by women as contributory factors to the poor utilization of the maternity facilities for deliveries. This corroborates the findings of a review that revealed the culture regarding childbirth as a natural process that does not require interventions as one of the factors influencing hospital deliveries negatively [30].

The women also highlighted the importance of the provision of free maternity services, which encouraged utilization and enabled access to evidenced based facility-based delivery. This finding is congruent with the findings of a Kenyan study, which reported an increase in the number of deliveries and antenatal attendance by 26.8% and 16.2% in county referral hospitals as a result of free delivery policy [31]. In the current study, women further indicated the availability of both material and human resources and a good interpersonal relationship with healthcare workers as facilitators of facility-based delivery. They also revealed that negative attitude of nurses poses a hindrance to utilization. These findings were echoing the results of a review that indicated the availability of resources and being treated respectfully as the promoter of hospital delivery [32]. Therefore, developing trust between communities and facilities will be essential in increasing FBD, but this will only be achieved if facilities provide good quality care [33].

## 5. Conclusions

This study provides a valuable contribution to the literature. This study attempts to address the challenges faced by pregnant women in accepting FBD, in order to facilitate the acceptance of FBD, thus reducing maternal and neonatal mortality rates and improving maternal and child health. Results showed several factors associated with the failure to use institutional delivery services, such as long distance from the health care facility, lack of transport, lack of transport fare, shortages of skilled staff, failure to disclose pregnancy, cultural and religious beliefs, and staff attitudes. Provision of free maternal services alone is not enough to increase FBD to the desired level, but challenges must be addressed to remove barriers to acceptance of FBD.

## Figures and Tables

**Table 1 ijerph-20-06535-t001:** Description of the demographic characteristics of participants.

No	Age in Years	Marital Status	Parity	Level of Education	Name of Health Facility
1	25	Single	Para 1, Gravida 2	Grade 11	CHC—A
2	27	Single	Para 0, Gravida 2	Grade 12	CHC—A
3	31	Married	Para 2 Gravida 3	Grade 10	CHC—A
4	32	Single	Para 1 Gravida 3	Grade 12	CHC—A
5	27	Single	Para 1, Gravida 2	Grade 8	CHC—A
6	33	Single	Para 2, Gravida 3	Grade 10	CHC—A
7	26	Single	Para 2 Gravida 3	Grade 12	CHC—A
8	31	Single	Para 1 Gravida 2	Grade 10	CHC—A
9	37	Married	Para 3 Gravida 4	Grade 6	CHC—A
10	36	Married	Para 3 Gravida 4	Grade 6	CHC—B
11	35	Single	Para 2 Gravida 3	Grade 12	CHC—B
12	25	Married	Para 1 Gravida 2	Grade 12	CHC—B
13	31	Single	Para 1 Gravida 2	Grade 9	CHC—B
14	34	Married	Para 2 Gravida 3	Grade 12	CHC—B
The mean age of the pregnant women was 30.7					

## Data Availability

The data presented in this study are available upon request from the corresponding author. The data are not publicly available due to privacy restriction.

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
