# Peer review of "Pregnant Women’s Views Regarding Maternity Facility-Based Delivery at Primary Health Care Facilities in the Province of KwaZulu-Natal in South Africa"

_ijerph, 2023, doi:10.3390/ijerph20156535_

Round 1

Reviewer 1 Report

Reviewers Report:

Title: Knowledge of pregnant women regarding maternity facility-based delivery at primary health care facilities in the province of KwaZulu-Natal in South Africa

Thank you for the opportunity to review this paper, which will be of interest to those in this field.

Title: suggestion ‘Women’s views on facilitators and challenges to utilisation of facility-based maternity care’

Abstract: line 17-18 “Information was collected from six groups of pregnant women who had delivered babies at the antenatal care facilities in the past years.”

Do you mean ‘ primary healthcare facilities’?

Introduction: Line 39-41 According to these findings, maternal and child healthcare is not reaching everyone in urban sub-Saharan Africa including South Africa. 

Do you mean rural areas?

In line 70-71 you stated that ‘…yet less than 55% of these women, mostly in the rural communities,’

Lines 42-45 “…majority of deaths occurred among those whose educational status was unspecified (46.2%) and lowest among those with no education, primary school education and university education” [10]

Any explanation as to why death rate is low among those with no education?

Line 46 “According to Yaya et al. [7], women in rural areas in both countries regarded facility…”

Which countries?

Line 75 BANC- write in full for the first time.

Material and methods:

It would good to state the study methodology (e.g. qualitative study) and rationale for the choice.

Lines 94-96 – It is obvious which one is site A and B. It would be better to say ‘the sites are referred to CHC A and B’.  

Participants and Data generation:

How were the participants recruited? Data collection method? When did data collection occur? What was the data collection location? It would be good to provide examples of the questions that guided the group discussions. You mentioned in the abstract (lines 16-18) that Focus group discussions were used to collect data (six groups). Explain this in detail.

Level of education Grade 12 and Grade 6 may not be clear to international readers.

Table 1: the participants have been linked to the sites and one of the sites is clearly identified while the other named CHC-O1, which contradicts your statement in lines 94-96. It would be good to be consistent.

Results

Safe Delivery: lines 169-170: you referred to compassionate and respectful care, perhaps this comment is relevant under 3.4.1 as demonstrated in the quotes in line 344 - 346

Lines 186-190 any quotes to show how the participants referred to caesarean section.

Line 215 and 220 ‘breach presentation’ should be ‘breech presentation.’

Line 216: you made reference to ‘level 1 facility’ – perhaps a brief overview if South African healthcare structure would be helpful for international audience.

Lines 315-327: Analysis on use of broken bottles for episiotomy (under ‘cultural belief’) is repeated in lines 362-370 (availability of logistics). Is this not a safety / resource issue? Perhaps it may be better to present it under ‘safe delivery’. 

Discussion: Lines 407-408 ‘These findings were 407 supported by Raymondville et al…’

Your finding is similar to or support previous studies.

Overall the discussion could be supported with more literature.

Conclusion: it would be good to highlight the key findings here.

Reviewer 2 Report

Dear authors,

First of all, I would like to congratulate you for approaching such a challenging subject, and for managing to interview these women and succeeding in presenting their raw opinion regarding maternity-based delivery at primary healthcare facilities in Africa. Although the subject is intriguing and it may consist in a valuable scientific asset it needs improvement.

The article needs moderate English revision – there are plenty of passive voice misuse and not so fortunate word choices. The article has no statistic section at all. You should at least try to compare women’s responses depending on marital status or rural/urban provenance. It would be better if you insert in the article the questionnaire you used for the article.

The title needs some rephrasing. First of all delete “Title:”, and second, if you accept suggestions perhaps “Pregnant women’s opinion regarding maternity-based delivery at primary healthcare facilities in the province of KwaZulu-Natal in South Africa”.

Abstract section

Some commas missing in the abstract

Row 11 - before “neonatal and maternal”

Row 13 before “therefore” not after

Row 15 – perhaps “This study aimed” sound better

Row 17 – “the pregnant women” without the preposition

Row 19 – services not service

Introduction

Rows 29-30 – perhaps without “the period of” would be better; just “throughout pregnancy”

Row 31 – a comma before “and”

Row 34 – health without the preposition, without the “the”

Row 37 – utilization not utilization

Row 42 – a comma before “and”

Row 47 – the same, a comma before “and”

Row 61 - healthcare not “health care”

Row 65 – “sub-region” or “sub-Saharan region”

Row 66 “the scourge of unsupervised births” add the preposition

Row 67 – in particular needs commas before and after

Row 77 – ANC services without the comma                             

Rows 83-84 – rephrase for better understanding

Row 84 – “the” poor quality

Row 85 – comma before therefore

Row 87 – “the” global reduction

Row 89 – just “to” not “in order to”

Materials and methods:

Row 93 – located “in”

Row 94 –“the anonymity”

Row 95 – without “for the purpose of” sounds better

Participants and Data Generation perhaps data collection would be better and also rephrase the first sentence that follows “A total of 14 participants from the health facilities under study, nine pregnant women from Hlengisizwe CHC-01 and five pregnant women from uMsunduzi Clinic-02.”

Hlengisizwe CHC-01 and five pregnant women from uMsunduzi Clinic-02 – the first mentioned classification in the paragraph above “Participants and Data Generation” tha classification was CHC A and B – why is it with 01 and 02 now? Why 01 and not just 1? And this Hlengisizwe region is from what part? It was not described before in the first two “uMzinyathi District and uThukela District”.

Table 1 – instead of “p” you should probably use “No”

Row 109 - All 108 pregnant women were confirmed pregnant - rephrase

Rephrase into “Their highest level of education was 12 Grade, with 6 Grade being the lowest level of education.”

Durban University of 114 Technology’s Institutional Research Ethics Committee (131/19) – the date is missing

Approval was also 115 granted by the Province of KZN Department of Health. – number and date of the approval

Row 119 – “the” ethics; add the preposition

Results:

Row 144 – “healthcare”

Row 145 – capitalize the first letter in the sentence

Row 146 – “within reach”

Row 147 - “First-level”

Row 156 – “a” robust

Rows 164-167 – other format, with Italics, than the ones used before for mentioning patients comments – standardize the manner you present them

Safe-havens without the hyphen

Row 173 – “illustrated in” not “from”

Row 180 – increase not increases

Row 181 – “the early” with preposition

Row 182 – “the proper” with preposition

“Pregnant women further noted that it is exciting that all pregnant women” – rephrase

Roe 192 – without “further”

Row 205 – “the government” with preposition

Row 206 – supports not support

Row 213 – reported – a ”t” missing

Row 217 – rephrase

Row 227 - Capricious weather – instead of “rarely weather”

Row 227 - Without and before letter f

Row 230 – “the majority of population”

Row 233 – most of “the” women

Row 234 – without the preposition before “rural areas”

Row 235 – without “further”

Row 243 – due to “the” long distance

Row 244 – same comment “the long”

Row 245 – travelling – an “l” missing

Row 250 – only health without the preposition

Row 252 – consider changing “few staff” with “few healthcare providers”

Row 257 – “the” lack of staff; “according to her “ a comma missing after the word her

Change “at the postnatal” into “in the postnatal”

Row 259 – “her patronage of facility-based delivery” - rephrase

Row 260 – describes with an s at the end

Row 265 “a” lack

Row 268 – undergone not underwent

Row 298 – paragraph alignment missing

Row 298 – healthcare not health care

Row 300 – an extra “l” missing in travelling

Row 309 – to produce, not producing

Row 315 – alignment for new paragraph? And worthnoting with a space between worth and noting

Row 319 – relevant equipment was not were

Row 323 – were brought to health facilities not to “the” health facilities

Row 336 – illustrated by not from

Row 340 new paragraph alignment missing

Row 342 – the comma before and after therefore

Row 351 – illustrated by

Row 356 “The availability”; a coma before especially; when the weather not were the weather

Discussion:

Row 377 – emergencies not emergency

Row 378 – “the” treatment

Row 383 – “an” intermediation

Rows 385-393 – repetitive, same as the paragraph above

Row 395 – change “high price” to higher costs

Row 398 – “the cost”

Row 401 without “with”

Row 402 – rephrase “late booking for ANC”

Row 403 – backgrounds

Healthcare not health care

Row 416 – childbirth

The article needs a stronger discussion section. You need to include all the significant research about this topic in literature. There are only few data mentioned. Due to the low number of participants you need to make your work valuable and compensate with up to date extensive discussions.

Conclusions:

Row 419 – “the facility based” without the preposition

Row 421 – improving not improve

“but strong efforts must be escalated” – rephrase

“facility-based delivery, so as to facilitate acceptance of facility-based” - rephrase because it is to repetitive

Row 431 – close the parenthesis after the ethics reference number.

moderate English revision needed

Reviewer 3 Report

The paper was well written, however there are two major issues that I have with the overall research approach. First, the research question and choice of participants is not congruent. Why were pregnant women selected as participants for the qualitative focus groups? It would seem that the participants should have been women who had birthed either in or out of a facility.  If you could follow up to find out where the participants gave birth and then compare their responses it would potentially be an interesting paper. The other issue is there is no methodology for the analysis of the qualitative data. If you are able to determine where participants gave birth, then you could do qualitative comparison (compare the qualitative results from women who gave birth in the facility to those that birthed outside to explore attitudes toward facility births) Otherwise I don't think it really adds to the literature. Here is a scoping review that describes qualitative comparison https://journals.sagepub.com/doi/abs/10.1177/1049732318807208

Round 2

Reviewer 1 Report

The suggestions have been addressed.

Just an observation in line119: should read - All pregnant women who participated in this study.

Line 120: It was deemed necessary to assess these pregnant women views about their views, (delete their views)

142: missing word "what do you do you know about facility-based delivery"?

Minor editing as noted above

Author Response

Point 1: Just an observation in line119: should read – All pregnant women who participated in this study.

Response 1: Thank you. Amendment has been made.

Point 2: Line 120: It was deemed necessary to assess these pregnant women views about their views, (delete their views).

Response 2: Thank you. Amendment has been made.

Point 3: 142: missing word "what do you do you know about facility-based delivery"?

Response 3: Thank you. Amendment has been made.

Reviewer 2 Report

Congratulations on improving your work.
